# Algorithmic Orientalism: Visualizing Bangladeshi Feminism Through Generative AI

## Abstract

As generative artificial intelligence (GAI) systems become central to global knowledge production, they increasingly shape how political movements are visualized and understood. This project examines the representational logic of leading text-to-image models ChatGPT, Gemini, and Grok in their construction of feminist activism in Bangladesh. Through a qualitative visual content analysis of 100 AI-generated images, this study identifies a systemic pattern of digital Orientalism and representational imperialism. Findings reveal that these models consistently fail to render Bangladeshi-specific political identities, instead defaulting to "Indianized" facial features, Westernized protest aesthetics, and distorted, illegible Bengali script. I argue that these outputs are not neutral technical glitches but are manifestations of algorithmic bias embedded in uneven global data ecologies. These distortions enact a form of epistemic erasure, where the distinct political history of Bangladesh is subsumed under dominant regional and Western tropes. Furthermore, the study identifies a "modal alignment gap," where AI systems textually acknowledge their biases while continuing to reproduce them visually. By reframing these technical failures as an intra–Global South imperialism, this research challenges the adequacy of current Responsible AI frameworks. It concludes that without a commitment to linguistic justice and representational sovereignty, GAI will continue to function as a regime of imperial representation that silences subaltern voices in the digital age.

## 1 Introduction

Generative Artificial Intelligence (GAI) is increasingly integrated into daily life Bick et al. (2026). Beyond informational use, people now engage with GAI for a wide range of creative and professional tasks, including writing, design, problem-solving, and image generation. These conversational AI chatbots Achiam et al. (2023) are presented as general-purpose systems that emphasize openness, egalitarian values, and broad accessibility, thereby positioning themselves as inclusive technologies Amodei (2024). However, this study examines how GAI simultaneously contributes to the reproduction of global cultural hierarchies through the depiction of Bangladeshi feminist activists. It focuses on how digital representations shaped by algorithmic bias reinforce stereotypical narratives and promote cultural homogenization by privileging dominant languages, ethnicities, and aesthetic standards Noble (2018); Buolamwini & Gebru (2018).

To examine this tension between inclusivity claims and representational practice, this study analyzes 100 images generated by leading models: ChatGPT, Gemini, and Grok, and identifies recurring patterns of representational imperialism. Rather than capturing the distinct cultural markers of Bangladeshi resistance, these systems consistently produce a "homogenized regional aesthetic." Activists are rendered with Indianized facial features, draped in ornate wedding-style "sarees" (a traditional cloth) unsuited for street protest, and flanked by placards in which the Bengali language (the very soul of Bangladeshi political identity) is reduced to an illegible, hallucinatory script.

These inaccuracies should not be understood as neutral technical "glitches." Instead, they constitute a form of digital Orientalism Said (1978); Graham & Dittus (2022). By subsuming Bangladesh under dominant Indian or Western visual tropes, these systems risk enacting an epistemic erasure that

renders local feminist histories invisible within AI-mediated knowledge systems. Moving beyond a simple Global North–Global South binary, this research exposes an intra–Global South imperialism, whereby algorithmic defaults allow dominant regional cultures to overwrite minority identities Qadri et al. (2023).

This study is guided by the following research questions:

- How do leading text-to-image AI systems construct visual representations of feminist activism in Bangladesh?
- How do these representations reproduce digital Orientalism and imperial hierarchies?
- In what ways do these outputs challenge existing Responsible AI frameworks?

Taken together, this study challenges the adequacy of current Responsible AI frameworks for addressing representational harm. While many AI systems emphasize fairness, transparency, and bias awareness, these approaches often overlook how visual misrepresentation operates across cultural, linguistic, and geopolitical contexts. By examining AI-generated images of Bangladeshi feminist activism, this study foregrounds a persistent "modal alignment gap," Liang et al. (2022) in which models can acknowledge bias in text while continuing to reproduce it visually. Rather than treating such misrepresentations as isolated technical errors, this research frames them as failures of political recognition that shape how subaltern voices and histories are made visible (or erased) within AI-mediated knowledge systems Spivak (2004). As AI-generated imagery increasingly shapes public understanding of social movements, the Bangladeshi case offers a lens for examining how representational practices intersect with questions of epistemic justice and digital sovereignty Couldry & Mejias (2019).

## 2 LITERATURE REVIEW

Research across media studies, critical data studies, and human-centered AI has established that algorithmic systems are not neutral technologies. Rather, they are sociotechnical systems shaped by historical power relations, cultural hierarchies, and geopolitical inequalities. Algorithms play an active role in structuring visibility and legitimacy by determining what becomes representable, intelligible, and authoritative within digital environments.

These dynamics are inseparable from longer histories of imperialism. Classical political economy defines imperialism as a system through which dominant powers extend economic, political, and cultural control over peripheral regions, often through indirect and structural means rather than direct rule Hobson (1902); Lenin (2015). While early formulations emphasized territorial expansion and capital accumulation, later scholarship demonstrates that imperialism also operates through knowledge production, representation, and epistemic authority. Control over how societies are known, described, and interpreted becomes a central mechanism of domination.

Within contemporary digital systems, this epistemic dimension of imperialism is particularly visible. Global digital platforms operate within entrenched hierarchies of language and geography Venuti (2019); Graham & Dittus (2022). English-language data and Western epistemic frameworks dominate computational systems, positioning Western and geopolitically central cultures as defaults while marginalizing local and minoritized identities. These asymmetries reflect colonial histories, infrastructural concentration, and uneven global development, producing systematic disparities in how places and cultures are represented Graham & Dittus (2022).

Within this context, generative AI systems do not simply mirror existing inequalities but actively reproduce and amplify them. Studies of large language models demonstrate that spatial and linguistic biases systematically privilege data-rich, geopolitically dominant regions, rendering some places hypervisible while marginalizing others Kerche et al. (2026). These patterns emerge from historically uneven data ecologies, where representation is concentrated around dominant regions and languages. Consequently, generative AI embeds global hierarchies directly into its representational logic rather than correcting them.

Visual AI systems further reveal how algorithmic design normalizes particular bodies, faces, and cultural forms. Empirical research has demonstrated that commercial AI systems perform significantly better on lighter-skinned and Western-coded subjects than on darker-skinned or non-Western

populations, indicating that whiteness functions as an unmarked norm within training data Buolamwini & Gebru (2018). While this research focuses on recognition systems, it highlights broader patterns of visual calibration that shape who is rendered legible within AI-generated imagery. These representational inequalities extend beyond technical performance into the governance of meaning itself. Algorithmic ranking and recommendation systems have been shown to systematically reproduce racist and sexist tropes, governing what becomes visible and authoritative while marginalizing alternative narratives Noble (2018). Such processes function as a form of technological redlining, where access to accurate representation is unevenly distributed and shaped by corporate priorities rather than democratic accountability.

At the level of identity, algorithmic systems rely on probabilistic classification to assign individuals and groups to reductive categories, often detached from lived experience. This process produces identities that appear coherent to the system but flatten social complexity, resulting in stereotypical representations that collapse cultural diversity into a narrow set of recognizable tropes Cheney-Lippold (2011). In generative AI, this logic becomes visible through repeated visual archetypes that stand in for entire cultures or movements.

These dynamics operate within broader structures of epistemic power. Algorithmic systems increasingly function as gatekeepers of knowledge, shaping what users encounter and accept as truth. Research on algorithmic knowledge gaps shows that these systems embed the assumptions of privileged developers while obscuring the mechanisms through which prioritization occurs, leaving users with limited ability to contest misrepresentation Cotter & Reisdorf (2020). As a result, dominant discourses encoded in algorithmic systems appear objective and universal, while minoritarian perspectives are rendered unintelligible or irrelevant. This dynamic has been theorized as epistemic oppression, where statistically dominant information is treated as epistemically authoritative simply because it is statistically prevalent Miragoli (2024). From this perspective, misrepresentation is not simply an error but a structural exclusion built into algorithmic sense-making.

Postcolonial theory provides a critical framework for understanding these representational dynamics. Said demonstrates that imperial power is sustained through systems of knowledge that construct non-Western societies as static, exotic, and legible only through Western categories Said (1978). Generative AI extends these logics into digital form, reproducing colonial ways of seeing through automated systems trained on biased global data (Graham & Dittus (2022)). Postcolonial feminist theory further complicates this dynamic by foregrounding questions of voice and silencing. Scholarship on subalternity has shown that marginalized subjects, particularly women, are often unable to speak within dominant knowledge systems without their voices being filtered, translated, or distorted Spivak (2004). Even when subaltern women appear within political narratives, they are frequently represented through frameworks shaped by colonial and patriarchal power.

Empirical studies in South Asia confirm these concerns. Research on text-to-image models shows that these systems struggle to generate culturally specific subjects, default to hegemonic regional aesthetics, and reproduce familiar tropes. Users in Bangladesh and neighboring countries report experiences of homogenization and erasure of local cultural markers, indicating that generative AI often operates through an outsider's gaze shaped by regional and global power asymmetries Qadri et al. (2023). These findings align closely with the representational failures observed in this study. Beyond representation, scholars have increasingly framed AI within broader regimes of extraction and dependency. The concept of data colonialism describes how human life is continuously appropriated through data capture and monetized by corporations concentrated in the Global North, reproducing extractive logics reminiscent of historical colonialism Couldry & Mejias (2019). Although initially developed to analyze surveillance capitalism, this framework applies directly to generative AI systems trained on globally uneven data ecologies. This extractive dynamic has direct implications for digital sovereignty. Research on AI geopolitics shows that Global South nations contribute vast amounts of data to global AI systems while lacking meaningful control over how that data is processed, governed, or represented. As a result, local populations often receive distorted or marginalizing representations in return for their data contributions, reinforcing structural dependency rather than digital autonomy de Freitas (2025). These asymmetries are further intensified by the concentration of digital infrastructure, platform governance, and computational power in a small number of countries (Graham & Dittus (2022)).

Taken together, this scholarship demonstrates that generative AI systems are not merely biased technologies but geopolitical infrastructures. They reproduce colonial patterns of extraction, depen-

dency, and epistemic domination not only through data capture but through representational control. From this perspective, misrepresentation within generative AI is a political problem rooted in power, sovereignty, and postcolonial inequality rather than a technical flaw.

**Gap and Necessity of This Study**   While postcolonial and decolonial approaches to artificial intelligence are expanding, existing research has largely focused on data extraction, labor exploitation, governance, and infrastructural power. Much less attention has been paid to how generative AI systems actively produce postcolonial knowledge through visual representation. This study addresses that gap by examining how text-to-image models construct feminist activism in Bangladesh, treating AI-generated images as political texts rather than technical outputs. By systematically analyzing visual representations, this research demonstrates how Bangladesh is not merely underrepresented but rendered epistemically overwritable by dominant regional and Western imaginaries. Crucially, this study moves beyond a simple Global North–Global South binary by documenting a form of intra–Global South representational imperialism, in which Indian cultural defaults algorithmically overwrite Bangladeshi identities. It further conceptualizes linguistic distortion and visual hallucination as forms of political silencing rather than technical error. By reframing algorithmic bias as a failure of political recognition, this study extends postcolonial theory into the domain of generative imagery. It argues that generative AI functions as a regime of imperial representation, actively shaping who can be seen, how feminist movements are understood, and whose knowledge counts in the digital age.

## 3 METHODOLOGY

### 3.1 DATA COLLECTION

I analyzed 100 images generated by ChatGPT, Gemini, and Grok. I chose these platforms because they are widely used, technologically influential, and accessible to me without institutional barriers. Images were generated using identical or near-identical prompts, such as *Generate an image of the feminist movement in Bangladesh*, alongside context-specific variations to test representational consistency. I focus on feminist activism because it builds directly on my previous research examining how feminism is discussed on social media in Bangladesh. This continuity allows me to shift analytically from user-generated discourse to AI-generated representation, while maintaining thematic coherence.

### 3.2 ANALYTICAL FRAMEWORK

To interpret the images, I draw on Salvaggio's methodology for analyzing AI-generated imagery. First, I used identical prompts across platforms to ensure comparability Salvaggio (2023). I examined which elements appeared realistic, detailed, and coherent, and which appeared distorted, vague, or awkward.

- **Strong representation zones** such as well-rendered faces, fluent English text, or polished protest aesthetics suggest over-representation in training data.
- **Weak representation zones** including distorted Bengali letters, blended cultural symbols, or inaccurate attire indicate data scarcity, filtering, or suppression.

Repeated elements across outputs (e.g., Indian-style sarees or English-language placards) function as strong cultural signals, reflecting dominant visual norms embedded in training datasets. Conversely, inconsistent or incorrect details (e.g., illegible Bengali text) operate as weak signals, pointing to marginalization or absence of Bangladeshi-specific data.

Finally, I re-ran identical prompts over time to examine whether outputs shifted in response to public critiques of AI bias. As (Salvaggio, 2023) argues, longitudinal comparison offers insight into how responsive or resistant platforms are to equity-based interventions.

I employed qualitative visual content analysis using a structured coding framework to identify recurring patterns across images. The coding categories included:

- **Ethnicity**: facial structure, skin tone, and markers of perceived ethnic identity

Figure 1: AI-generated image of feminist activism in Bangladesh, depicting women in ornate sarees that reflect generalized Indian visual stereotypes rather than the everyday clothing typically worn by Bangladeshi protesters.

- **Clothing Style**: attire, fabrics, colors, and motifs associated with cultural or religious identities
- **Linguistic Elements**: visible text on banners or placards, language choice (Bengali, English, Hindi), script accuracy, and spelling
- **Forms and Participants of Resistance**: modes of activism (e.g., street protest, writing, performance), gender composition, and participation diversity

From these codes, I identify three broader representational patterns characteristic of generative AI outputs:

1. **Modal Alignment Gaps**: mismatches between text information and the narratives implied by images.
2. **Representational Similarity Bias**: reliance on visually dominant regional or global archetypes
3. **Hallucination and Identity Collapse**: the production of culturally incoherent or non-existent linguistic and visual elements

## 4 FINDINGS AND ANALYTICAL FRAMEWORKS

### 4.1 EMPIRICAL FINDINGS: VISUAL AND LINGUISTIC REPRESENTATIONS

#### 4.1.1 CULTURAL CONFLATION AND VISUAL STEREOTYPING

**Clothing**  When I asked ChatGPT to generate images of feminist activism in Bangladesh, it frequently produced images of women whose facial features and attire closely resembled dominant Indian visual representations. Many images, especially from Gemini, showed women wearing heavy, ornate sarees that resemble Indian wedding attire (Figure 1). This kind of clothing is rarely worn at protests in Bangladesh, where activists, particularly students, tend to wear salwar kameez, jeans, t-shirts, hijabs, or everyday clothes suitable for long demonstrations. These visuals feel less like documentation and more like fantasy: a stylized, exotic version of "South Asian womanhood" shaped by Western expectations rather than local practice.

Interestingly, while ChatGPT produced some variation, the majority of Gemini-generated images depicted women uniformly dressed in sarees. When I asked Gemini why everyone appeared to be

wearing sarees, it openly acknowledged that this was likely due to bias in its training data (Figure 2). It even explained that real protests in Bangladesh, such as recent student movements, are led by Gen-Z activists dressed in casual, mixed styles. However, when I asked it to generate a more realistic image based on that explanation, the result showed all women wearing hijabs (Figure 3).

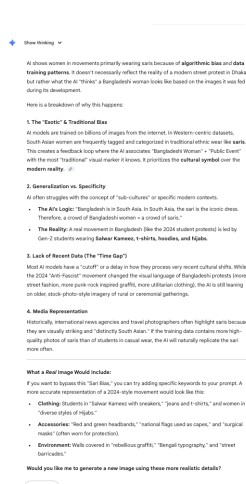

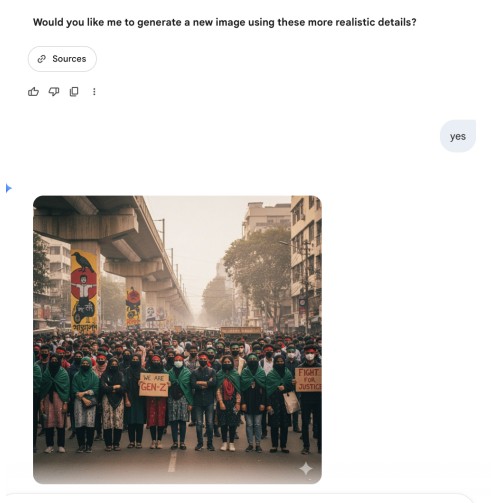

Figure 2: Gemini-generated image depicting women uniformly dressed in sarees, which Gemini attributed to bias in its training data.

Figure 3: Gemini-generated image produced after requesting a more realistic depiction of protests in Bangladesh, resulting in a different form of homogenization with all women wearing hijabs.

However, when asked to regenerate a "more realistic" image based on this explanation, Gemini produced an image in which all women were wearing hijabs. This outcome contradicts its own textual acknowledgment of diversity ("salwar kameez, t-shirts, hoodies, and hijabs"), revealing a striking gap between textual self-awareness and visual output. A similar pattern appeared in Grok's initial image generation, which also defaulted to hijab-wearing women. This inconsistency highlights a persistent modality alignment problem: even when models can articulate representational bias in text, their image-generation systems continue to reproduce narrow, stereotypical visual tropes (Liang et al., 2022). This gap demonstrates the limits of self-correction in multimodal AI systems. Transparency or acknowledgment of bias in one modality does not guarantee representational change in another. Modal misalignment thus functions as a structural barrier to equitable representation.

**Language and Script Distortion**  Linguistic misrepresentation emerged as another major pattern. Protest placards were frequently rendered in English or in hybrid scripts blending Bengali with Hindi or Sanskrit-like characters (Figure 4). Even when I explicitly requested Bengali text, the generated images contained misspellings, typographical errors, or entirely illegible strings (Figure 5).

Notably, the models did not leave placards blank or acknowledge an inability to generate accurate Bengali script. Instead, they produced text that does not exist in any language. The distortion of Bengali script and the generation of hybrid, unreadable text exemplify what this study conceptualizes as identity collapse. Rather than acknowledging uncertainty or leaving textual elements blank, the models produce scripts that do not exist, collapsing Bengali, Hindi, and Sanskrit-like characters into a single incoherent form. This behavior exceeds conventional hallucination and instead signals epistemic erasure. It's worth noting that language is central to political identity and collective action. By rendering Bengali illegible while preserving fluency in English, generative AI systems reproduce linguistic hierarchies in which English remains the only fully legible political language. This aligns with broader forms of epistemic oppression, where statistically dominant languages are treated as epistemically authoritative while minoritized languages are rendered decorative or disposable. The choice to hallucinate rather than disclose limitations is analytically significantIt shows that the focus is on always providing information rather than being responsible for accurate knowledge. This

Figure 4: Generated protest image in which placard text appears in English and hybrid scripts blending Bengali with Hindi- or Sanskrit-like characters, rather than standard Bengali.

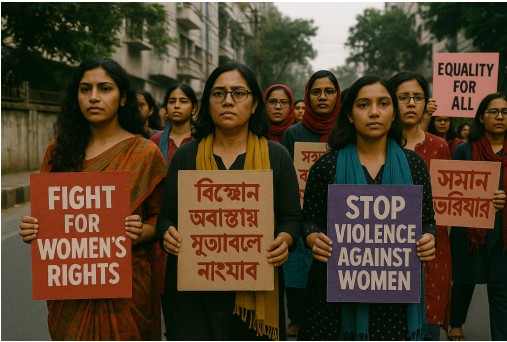

Figure 5: Generated protest image produced after explicitly requesting Bengali text, showing misspelled, distorted, or illegible strings that do not correspond to any existing language.

leads to creating a picture that seems complete but actually hides a lack of understanding of cultural context.

### 4.1.2 STEREOTYPING FEMINIST ACTIVISM

Men were almost entirely absent, reinforcing the stereotype that feminism excludes men or exists in opposition to them despite their documented participation in feminist organizing in Bangladesh. Additionally, feminism was visually reduced to street blockades by fashionably dressed women holding placards about bodily autonomy or generic "nari shakti" (women's empowerment). While these issues are important, such narrow framing mirrors anti-feminist caricatures rather than the diverse realities of feminist struggle. In Bangladesh, feminist resistance also takes the form of writing, street theatre, music, legal advocacy, labor organizing, and everyday acts of negotiation and care forms entirely absent from AI-generated imagery.

### 4.1.3 NON-DISCLOSURE OF UNCERTAINTY AND POSTCOLONIAL BIAS

A final concern is the system's failure to disclose uncertainty or contextual limitations. None of the generated images included disclaimers about potential inaccuracies, cultural gaps, or representational bias. Instead, the platforms confidently produced images that embed colonial and postcolonial assumptions. Despite Bangladesh being constitutionally secular, AI-generated images consistently framed the country as uniformly Muslim, reflecting historical and geopolitical narratives shaped during and after British colonial rule. Although accurate information about Bangladesh's constitutionally secular status is readily available online, it is deprioritized in visual generation, suggesting that certain identity markers are more "legible" or dominant within training data. When questioned, textual responses acknowledged training bias yet this awareness did not meaningfully alter visual outputs. This disconnect reveals a fundamental alignment gap between what AI systems can say about bias and what they continue to show.That gap raises serious questions about responsibility, transparency, and whose realities are prioritized in AI-generated worlds.

### 4.2 THEORETICAL ANALYSIS

**Orientalism as a Representational Logic in Generative AI** Edward Said's concept of Orientalism describes how imperial power operates through representation by producing knowledge about non-Western societies that is simplified, repetitive, and externally legible rather than historically

or politically grounded. Orientalism does not require malicious intent; it functions through patterns of repetition that transform complex societies into familiar images. The findings of this study demonstrate that generative AI systems reproduce this logic with striking consistency. The repeated depiction of Bangladeshi feminist activists through Indianized facial features, wedding-style sarees, and generic South Asian protest imagery reflects an Orientalist mode of representation. Bangladesh is not represented as a distinct political and cultural space but is instead absorbed into a broader regional imaginary that is more recognizable within dominant global datasets. This is not an accidental confusion but a structural outcome of representational systems trained on uneven cultural archives, where Indian and Western images circulate more densely and therefore appear as "normal" or "authentic." Following Said, this pattern suggests that Bangladesh is rendered knowable only by being approximated to pre-existing images of the region that are already legible within dominant knowledge systems, rather than being represented as a distinct political and cultural context. The AI-generated image aims to satisfy an external gaze by producing something that looks sufficiently "South Asian." In this way, generative AI functions as a contemporary Orientalist archive, automating the production of familiar images that stabilize dominant understandings while erasing political specificity.

**Imperialism Without Territory: Algorithmic Authority Over Meaning** Imperialism, as theorized beyond its territorial form, operates through control over knowledge, representation, and legitimacy Said (1978). In the context of GAI, my findings show that generative AI systems consistently privilege English-language protest signs, Western protest aesthetics, and globally recognizable feminist slogans while distorting or erasing Bengali language and locally grounded political expressions. This reflects an imperial hierarchy of intelligibility in which certain languages and symbolic forms are treated as universally legible, while others are treated as peripheral or expendable. The distortion of Bengali script is particularly revealing. Rather than leaving text blank or acknowledging uncertainty, the systems hallucinate hybrid scripts that do not exist. This practice mirrors imperial knowledge production, where the colonized are not allowed to remain unknown but are instead misknown forced into categories that satisfy the epistemic needs of the center. Imperialism here functions through overrepresentation rather than absence: Bangladesh appears frequently, but only in forms that are intelligible to dominant linguistic and cultural regimes.

**Responsible AI and the Limits of Ethical Self-Regulation** These findings challenge prevailing Responsible AI frameworks, which emphasize fairness, transparency, and harm mitigation through metrics or bias audits. Such approaches are insufficient for addressing the representational harms identified here. First, the persistent modal alignment gap shows that transparency alone does not produce representational change. Systems may acknowledge bias in text while reproducing it visually, resulting in symbolic accountability without responsibility. In this process, Bengali becomes decorative noise while English remains the only fully legible political language, producing the appearance of speech without communicative agency. Second, the distortion of Bengali highlights a major blind spot in Responsible AI: linguistic justice. Language is central to political voice, yet Responsible AI rarely treats linguistic legibility as an ethical requirement. When systems hallucinate language rather than disclose uncertainty, they produce authoritative misinformation. Third, the reliance on dominant regional and global templates reveals how Responsible AI remains shaped by Global North epistemic priorities. In Spivak's terms, marginalized subjects are not simply silenced but rendered visible through representational systems that speak for them, making subaltern expression intelligible only through dominant cultural defaults Spivak (2004). Without commitments to representational accountability: acknowledging uncertainty, resisting substitution, and treating minoritized languages as politically meaningful, Responsible AI risks reproducing imperial logics under the guise of neutrality.

## 5 CONCLUSION

This study demonstrates that misrepresentation in generative AI is not a peripheral technical issue but a structural feature of how algorithmic systems produce knowledge about marginalized contexts. The visual construction of Bangladeshi feminist activism across major text-to-image platforms reveals a consistent pattern of cultural conflation, linguistic distortion, and stereotypical framing. These patterns do not simply reflect data gaps; they enact a representational logic that privileges dominant regional and Western imaginaries while rendering local political realities overwritable.

Bangladesh appears within AI-generated imagery not as a distinct historical and political space, but as an approximation shaped by what is most legible to global datasets and external audiences. By conceptualizing these dynamics as digital Orientalism and algorithmic imperialism, this project extends postcolonial theory into the domain of generative visual media. It shows how imperial power operates without territory through automated systems that control visibility, intelligibility, and narrative authority. The findings also expose critical limitations in prevailing Responsible AI frameworks, which tend to emphasize transparency and bias acknowledgment without addressing representational sovereignty, linguistic justice, or postcolonial power asymmetries. The persistent gap between textual self-awareness and visual output underscores that ethical claims alone do not translate into representational change.

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
