# OpenReview forum: "Algorithmic Orientalism: Visualizing Bangladeshi Feminism Through Generative AI"
_ICLR.cc/2026/Workshop/AFAA — Submitted to AFAA 2026_

### Official Review · Reviewer_Ytih · 2026-02-12
**Review of "Algorithmic Orientalism: Visualizing Bangladeshi Feminism Through Generative AI"**

**Rating:** 1
**Confidence:** 5

**Summary:**

The author/s talk about one of the most crucial  ethical issues surrounding Text-to-Image (T2I) generation systems. The study provides an interesting yet well know insights on how such systems stereotype minority representations overconfidently. Specifically, the paper examines the visual construction of feminist activism in Bangladesh by assessing 100 images generated by commercial models like ChatGPT, Gemini and Grok. The authors claim that models default to "Indianized" aesthetics and fail to render Bengali script, framing these failures as "intra-Global South imperialism."

**Strengths:**

* The paper addresses an important topic regarding cultural representation in generative AI.
* The identification of the "modal alignment gap" (where the text acknowledges the bias but images do not) is an interesting observation.

**Weaknesses:**

**1. Very weak literature review**
A significant flaw in this work is its isolation from the existing body of literature on diffusion model bias. The authors present the stereotyping of minority demographics as a largely exploring theoretical issue, yet this has been rigorously quantified in prior prominent works. The failure to cite and engage with foundational studies, such as Luccioni et al. [1], undermines the paper's theoretical positioning. By relying heavily on sociotechnical terminology (e.g., "epistemic erasure") while ignoring the established technical state-of-the-art, the paper constructs a narrative that feels disconnected from the machine learning reality. It critiques models as "imperialist" agents without analyzing them through the lens of known training data biases or objective functions.

**2. Inadequate explanation of Audit Methodology**
The audit methodology lacks the transparency and rigor expected for a main track paper submission. The evaluation appears to be entirely subjective; there is no description of a structured observation protocol, nor was a user study conducted to establish inter-rater reliability. Furthermore, the analysis lacks a comparative baseline or "ground truth". The authors assert that the images are inaccurate but fail to provide visual examples of what a "correct" Bangladeshi protest or accurate Bengali placard should look like. Without these benchmarks, the claims of "distortion" rest solely on the authors' personal interpretation rather than reproducible fact.

**3. Speculative Claims & Technical Reality**
The paper leaps to strong sociotechnical conclusions, without investigating technical root causes. The inability of diffusion models to render complex, non-Latin scripts is a well-documented technical challenge [2], for which specific mitigation methods already exist [3]. A scientific approach would have assessed whether these failures stem from "intra-global South imperialism" or simply from data scarcity. For instance, a basic frequency analysis of Bengali tokens in public datasets like LAION-5B (now LAION-2B) [4] would likely provide a more rigorous, data-driven explanation for the observed performance degradation.

Ultimately, this submission presents sweeping claims that are not supported by the empirical evidence. The lack of a rigorous experimental setup, combined with a failure to engage with relevant technical literature, results in a paper that feels like a low-effort commentary rather than a scientific contribution. This is also evident from the way the results are presented in the paper like simply including the screenshot (low-image quality) of the chatbot screen (Figure 2) or swapped image captions (Figures 4&5). I therefore recommend a strong rejection.

[1] Luccioni, Sasha, et al. "Stable bias: Evaluating societal representations in diffusion models." Advances in Neural Information Processing Systems 36 (2023): 56338-56351.

[2] Bosheah, Z.; Bilicki, V. Challenges in Generating Accurate Text in Images: A Benchmark for Text-to-Image Models on Specialized Content. Appl. Sci. 2025, 15, 2274. https://doi.org/10.3390/app15052274.

[3] Yang, Y.; Gui, D.; Yuan, Y.; Liang, W.; Ding, H.; Hu, H.; Chen, K. GlyphControl: Glyph Conditional Control for Visual Text Generation. In Proceedings of the Advances in Neural Information Processing Systems 36 (NeurIPS 2023), New Orleans, LA, USA, 10–16 December 2023.

[4] Schuhmann, Christoph, et al. "Laion-5b: An open large-scale dataset for training next generation image-text models." Advances in neural information processing systems 35 (2022): 25278-25294.

---

### Official Review · Reviewer_gqWK · 2026-02-19
**ALGORITHMIC ORIENTALISM: VISUALIZING BANGLADESHI FEMINISM THROUGH GENERATIVE AI**

**Rating:** 3
**Confidence:** 5

**Summary:**

The paper investigates how leading text-to-image generative AI systems (ChatGPT, Gemini, and Grok) visually represent feminist activism in Bangladesh. Through a qualitative visual content analysis of 100 AI-generated images, it demonstrates that these systems systematically erase Bangladeshi-specific political, cultural, and linguistic identities, instead reproducing Indianized, Westernized, and homogenized visual tropes. The study introduces the concepts of algorithmic Orientalism, representational imperialism, and a modal alignment gap—where models acknowledge bias textually but continue to reproduce it visually. Its main contributions are (1) empirically documenting visual and linguistic misrepresentation of Bangladeshi feminism in generative AI, (2) reframing these failures as structural, postcolonial forms of epistemic erasure rather than technical glitches, and (3) arguing that current Responsible AI frameworks are inadequate without commitments to linguistic justice and representational sovereignty

**Strengths:**

1. Originality: Introduces algorithmic Orientalism and intra–Global South representational imperialism as novel lenses for analyzing generative visual AI.

2. Technical and Methodological Soundness: Uses a clear, systematic qualitative visual content analysis with consistent prompts across multiple leading models.

3. Conceptual Contribution: Identifies and clearly articulates the modal alignment gap between textual bias awareness and visual outputs.

4. Clarity and Coherence: Well-structured, theoretically grounded, and clearly written, making complex postcolonial arguments accessible to AI research audiences.

5. Potential Impact: Offers important implications for Responsible AI, especially around linguistic justice and visual representation, with relevance beyond the Bangladeshi case

**Weaknesses:**

1. Limited Scope: The analysis focuses on a single national and political context, which may constrain the generalizability of the findings.

2. Sample Size: While suitable for qualitative analysis, 100 images may limit claims about broader systemic behavior across models and prompts.

3. Methodological Subjectivity: Visual interpretation and coding rely heavily on the author’s qualitative judgment, with limited discussion of intercoder reliability or validation.

4. Model Coverage: Only three prominent platforms are examined, leaving out open-source or regionally trained models that could offer useful contrast.

5. Technical Depth: The paper does not empirically link observed representational failures to specific training data, architectures, or deployment choices, which may limit technical actionable insights.

---

### Official Review · Reviewer_u3Fp · 2026-02-21
**The paper offers an interesting postcolonial critique of AI visual bias in representing Bangladeshi feminist activism, but its core weakness lies in its thin, single-coder methodology that cannot support the sweeping systemic claims it makes.**

**Rating:** 2
**Confidence:** 5

**Summary:**

The paper "Algorithmic Orientalism: Visualizing Bangladeshi Feminism through Generative AI" looks at how three text-to-image models - ChatGPT, Gemini besides Grok - create pictures of feminist protest in Bangladesh. After reviewing 100 images, the study detects a repeated combination of "digital Orientalism" and %"representational imperialism", with each model unable to depict identifiable Bangladeshi subjects and instead falling back on "Indianized" faces, Western protest symbols and garbled Bengali text. One central result is the "modal alignment gap", where the software states in words that it carries bias but keeps showing the same bias in pictures. By treating those visual errors plus technical "glitches" as deliberate removal of knowledge, the authors recast them as "intra-Global South imperialism", instances where majority cultures replace minority identities inside the algorithm. The article questions the effectiveness of present Responsible AI guidelines, claiming they omit necessary safeguards for linguistic fairness and for the right of subaltern groups to control their own depiction online.

**Strengths:**

1. Addresses a genuinely underexplored area: the intersection of postcolonial theory, feminist studies, and generative AI visual representation.
2. The concept of "intra–Global South imperialism" is a meaningful theoretical contribution, pushing beyond the oversimplified Global North/South binary.
3. The disconnect between textual acknowledgment of bias and continued visual reproduction of stereotypes is a well-observed and potentially significant empirical phenomenon that deserves further study.
4. The paper engages substantively with relevant interdisciplinary literature, drawing appropriately on Said, Spivak, Noble, Buolamwini & Gebru, and data colonialism frameworks.
5. The focus on Bengali script distortion as a form of linguistic and epistemic harm is original and analytically compelling.

**Weaknesses:**

1. The sample size (100 images, methodology unclear on exact distribution across platforms) is too small to support the sweeping systemic claims made throughout.
2. There is a single coder conducting qualitative visual content analysis with no inter-rater reliability measures, raising serious validity concerns.
3. The study lacks transparency about prompt variations, the number of images per platform, and temporal data collection windows.
4. The paper conflates description with causation repeatedly inferring intentional "epistemic erasure" from what may be explainable through data imbalance and technical architecture alone.
5. Policy and practical recommendations are underdeveloped despite the paper's ambition to challenge Responsible AI frameworks.

---

### Meta-Review · Area_Chair_Jvzj · 2026-02-23

**Recommendation:** Reject
**Confidence:** 4

**Metareview:**

The paper tackles an important and current issue on intra–Global South imperialism as observed in generative AI image generation bias. All reviewers agree that the topic is important, well-motivated, and the STS literature context is very well-presented. The reviewers' concerns highlight the small study size and concerns about validity (e.g., through annotation).

Based on these concerns, the paper would benefit from major methodological improvements, and thus the recommendation is to reject. We encourage the authors to follow the reviewers' suggestions on increasing the sample size, running more cohesive experiments, and append the related work with foundational papers on model diffusion bias.

---

### Decision · Program_Chairs · 2026-03-02

Reject